# Impact of Implant Surface Material and Microscale Roughness on the Initial Attachment and Proliferation of Primary Human Gingival Fibroblasts

**DOI:** 10.3390/biology10050356

**Published:** 2021-04-22

**Authors:** Marco Aoqi Rausch, Hassan Shokoohi-Tabrizi, Christian Wehner, Benjamin E. Pippenger, Raphael S. Wagner, Christian Ulm, Andreas Moritz, Jiang Chen, Oleh Andrukhov

**Affiliations:** 1Competence Center for Periodontal Research, University Clinic of Dentistry, Medical University of Vienna, 1090 Vienna, Austria; marco.rausch@meduniwien.ac.at; 2Division of Conservative Dentistry and Periodontology, University Clinic of Dentistry, Medical University of Vienna, 1090 Vienna, Austria; hassan.shokoohi-tabrizi@meduniwien.ac.at (H.S.-T.); christian.wehner@meduniwien.ac.at (C.W.); andreas.moritz@meduniwien.ac.at (A.M.); 3Institut Straumann AG, 4052 Basel, Switzerland; benjamin.pippenger@straumann.com (B.E.P.); raphael.wagner@straumann.com (R.S.W.); 4Department of Periodontology, School of Dental Medicine, University of Bern, 3012 Bern, Switzerland; 5Division of Oral Surgery, University Clinic of Dentistry, Medical University of Vienna, 1090 Vienna, Austria; christian.ulm@meduniwien.ac.at; 6School and Hospital of Stomatology, Fujian Medical University, Fuzhou 350002, China

**Keywords:** implant surface, peri-implant soft tissue, zirconia, titanium, human gingival fibroblasts, cell attachment, focal adhesion

## Abstract

**Simple Summary:**

In recent years, zirconia dental implant systems, also known as “ceramic implants”, became an alternative to the commonly used titanium implants. The clinical success of dental implants depends on several factors, particularly on the soft tissue formation around the implant abutment, but the implant surfaces, which at most support this process, are still to be found. In the present study, we cultured human gingival cells on surfaces that were made from either titanium or zirconia. These surfaces were differently treated and, subsequently, had different roughness: some surfaces were machined and smooth, whereas other surfaces were sand-blasted and rough. The cells ability to attach and grow was slightly decreased by rougher surfaces, whereas no effect of implant material was observed. Furthermore, the expression of some proteins mediating cell attachment to the surface was strongly affected by the roughness and only marginally by the implant material. We concluded that the behavior of gingival cells is primarily influenced by surface roughness, whereas no apparent advantage of either material could be observed. This suggests the importance of surface makeup in relation to how soft tissue healing around the implant is promoted and might provide new approaches for future research.

**Abstract:**

Due to the rising demand for zirconia (Zr) based implant systems, it is important to understand the impact of Zr and titanium (Ti) implants and particularly their topography on soft tissue healing. As human gingival fibroblasts (hGFs) are the predominant cells in peri-implant soft tissue, we focused on examining the effect of implant material and surface roughness on hGFs’ initial attachment, growth and the expression of proteins involved in the focal adhesion. hGFs isolated from eight healthy donors were cultured on the following surfaces: smooth titanium machined surface (TiM), smooth zirconia machined surface (ZrM), moderately rough titanium surface (SLA), or moderately rough zirconia surface (ZLA) for up to 14 days. The initial attachment of hGFs was evaluated by scanning electron microscopy. Cell proliferation/viability was assessed by cell counting kit 8. Focal adhesion and cytoskeleton were visualized by a focal adhesion staining kit. The gene expression of focal adhesion kinase (FAK), α-smooth muscle actin (α-SMA), and integrin subunits ITG-β1, ITG-β4, ITG-α4, ITG-α5, ITG-α6, was evaluated by qPCR. Cell proliferation/viability was slightly decreased by moderately rough surfaces, whereas no effect of surface material was observed. Cell morphology was strikingly different between differently treated surfaces: on machined surfaces, cells had elongated morphology and were attached along the grooves, whereas on moderately rough surfaces, cells were randomly attached. Surface roughness had a more pronounced effect on the gene expression compared to the surface material. The expression of FAK, α-SMA, ITG-β4, ITG-α5, and ITG-α6 was enhanced by moderately rough surfaces compared to smooth surfaces. Within the limitations of this in vitro study, it can be concluded that the behavior of primary hGFs is primarily affected by surface structure, whereas no apparent advantage of Zr over Ti could be observed.

## 1. Introduction

Dental implants are considered as a reliable long-term therapy for tooth substitution in edentulous patients [1,2]. The success of implant therapy is largely dependent on osseointegration, the ability of dental implants to fuse with bone, which ensures the load-bearing and functionality [3]. In implant-supported restorations, timely soft tissue healing and quantity and quality of the peri-implant soft tissue are important factors that influence the esthetic outcome, such as the thickness, contour, and color of the peri-implant mucosa, as well as the extent of epithelial keratinization [4,5]. In addition, the color of the underlying root restorative materials, such as the implant abutment, also influences the aesthetics of the overlying gingiva [6]. The attachment of the gingiva to the implant serves as a biological seal that prevents the development of inflammatory peri-implant diseases [7,8]. Peri-implant soft tissue also differs from its natural counterpart around healthy teeth through distinctive collagen fiber orientation and less vascularization [9].

Nowadays, the majority of commercially available implants, while possibly differing in alloy composition and manufacturing methods, are made of Ti [10]. In-depth research and long-term clinical survival made titanium the gold standard for implant material [11]. In the last years, new implant materials, particularly zirconia (Zr), were introduced due to higher aesthetic demands [12]. Clinical data suggest that Ti and Zr implants are generally comparable in osseointegration [13], but there are still some reports implying the superiority of Ti material [14]. There is no consistent opinion regarding the effect of implant material on soft tissue regeneration [15]. One study reported advantages for Ti when it came to the initial stage of the regenerative process [16], another reported epithelial attachment to favor Zr [17]. Zr surface was also shown to reduce bacterial adhesion, which might indicate further benefits for implant healing [18].

Implant healing after the placement into the jaw and peri-implant tissue formation involves different cell types. Implant surface characteristics are an essential factor influencing the cell response. It is well established that a moderately rough surface promotes the response of bone-forming cells and osseointegration [19,20,21]. Additionally, surface hydrophilicity and nanoscale roughness are considered as the essential parameters influencing osseointegration [22]. Furthermore, several novel approaches such as surface coating with bioactive peptides and laser treatment to create the specific architecture arose within the last years [23,24,25]. The impact of surface characteristics on soft tissue is less investigated and somewhat unclear. A recent systematic review suggests that surface roughness and coating might be essential parameters for the interface between the implants and soft tissue [26].

Human gingival fibroblasts (hGFs) are the predominant cells in peri-implant soft tissue, and their proliferation and adhesion are of particular importance to peri-implant mucosa healing [27]. These cells play an important role in extracellular matrix formation and remodeling while also taking a crucial role in inflammatory and bacterial immune response [28,29]. Attachment of hGFs to the implant surfaces is mediated by integrins, the family of transmembrane proteins, which regulate various signaling pathways. Integrin subunits are an important part of the focal adhesion (FA), which is an assembly of numerous proteins. Focal adhesion kinase (FAK) is a non-receptor protein tyrosine kinase, which is involved in the signal transduction from integrin complexes and activation of intracellular signaling pathways as well as other biological functions such as control of cell motility, proliferation, and migration [30]. The maturation and functionality of FA are closely related to the expression of α-smooth muscle actin (α-SMA) [31]. α-SMA is associated with myofibroblasts, a contractile fibroblasts phenotype, which plays an important role in tissue healing, matrix contraction, and remodeling [32].

Only a few studies dealt with the impact of implant surface material and structure on the response of hGFs. Microscale roughness is one of the key parameters influencing the cellular response. The effect of surface texture on fibroblasts is still poorly understood. Both implant material and surface topography seem to influence hGFs. Oates et al. reported that Ti surface roughness alters cellular morphology but has a limited effect on integrin expression [33]. Yamano et al. demonstrated that different surface materials and topographies might induce a distinct hGF morphology, proliferation, and in particular contrasting gene expression response of various integrin biomarkers [34]. Gómez-Florit et al. showed that moderately rough surfaces increase α-SMA expression and affect cell morphology in hGFs. Our recent study showed that implant surface roughness and, to a lesser extent, the material has a significant effect on the response of hGFs under inflammatory conditions [35]. However, the impact of implant material and roughness on the attachment and focal adhesion characteristics of hGFs is not entirely clear.

In the present study, we focused on examining the effect of implant material and surface roughness on the initial attachment and growth of hGFs. The initial attachment of hGFs was evaluated by scanning electron microscopy. Cell proliferation/viability was assessed based on cell metabolic activity. The gene expression of focal adhesion proteins FAK, α-SMA, and integrin subunits ITG-β1, ITG-β4, ITG-α4, ITG-α5, ITG-α6 was evaluated by qPCR. Additionally, focal adhesion was visualized using specific fluorescence staining.

## 2. Materials and Methods

### 2.1. Experimental Implant Discs

Four different implant surfaces with a dimension of 15 × 1 mm were used in the present study: a smooth machined Ti surface (TiM); a smooth machined ZrO_2_ surface (ZrM); a moderately rough, sand-blasted, large grain acid-etched Ti surface (SLA) and a moderately rough ZrO_2_ surface (ZLA). The samples were provided by the Straumann Institute (Basel, Switzerland). A detailed protocol of disks preparation and their characteristics in terms of roughness and wettability are provided in our previous study [35].

### 2.2. Cell Culture

Human gingival fibroblasts (hGFs) were isolated from healthy gingival tissues obtained from the extracted third molars of 8 healthy donors similar to previously described methods [35,36,37]. Patients were informed prior to extraction about the procedure and protocols and gave their written consent. The study was performed in line with the Declaration of Helsinki and “Good Scientific Practice” guidelines of the Medical University of Vienna. The study protocol was approved by the Ethics Committee of the Medical University of Vienna (EK Nr: 1079/2019). After extraction, teeth were placed in sterile vials containing Dulbecco’s Modified Eagle’s Medium (DMEM; Sigma-Aldrich^®^, St. Louis, MO, USA) nutrient medium supplemented with 10% fetal bovine serum (FCS; Sigma-Aldrich^®^, St. Louis, MO, USA), 100 U/mL penicillin and 50 µg/mL streptomycin (PS, Gibco^®^; Life Technologies, Carlsbad, CA, USA). To isolate hGFs, gingival tissue surrounding the tooth crown was cut out, minced, and distributed on Petri dishes in DMEM. Every 3 days, the medium was removed and replaced with a fresh portion of DMEM, and the growth of cells from the tissue pieces was observed under the light microscope. Cells in the passages between the 3rd and 6th were used in the experiments.

### 2.3. Experimental Protocol

Different discs were placed in a standard 24-well cell culture plate. After determining the cell count, hGFs cells were seeded on different implant surfaces in 1 mL of fully supplemented DMEM at a density of 2 × 10^4^ cells/well for proliferation/viability assays, scanning electron microscopy, and focal adhesion staining (SEM) or 1 × 10^5^ cells/well for qPCR. Cells seeded on tissue culture plastic at a similar density served as control. Cells were cultured at 37 °C, 5% CO_2_, and 95% humidity for up to 14 days. Each third day, the 500 µL of culture medium was replaced with 500 µL of fresh fully supplemented DMEM.

### 2.4. Cell Proliferation/Viability

Cell proliferation/viability was determined by cell counting kit 8 (CCK-8, Dojindo Molecular Technologies Inc., Gaithersburg, MD, USA). After 2 and 7 days, 100 µL of CCK-8 reagent solution was pipetted into each well, and cells were incubated at 37 °C for 2 h. Afterward, 100 µL of the solution was transferred into a new 96-well plate, and absorbance at 450 nm was measured using the Synergy HTX Multi-Detection Reader (BioTek Instruments, Winooski, VT, USA).

### 2.5. Scanning Electron Microscopy Analysis

The morphology and microstructure of hGFs grown on different implant surfaces were analyzed using scanning electron microscopy (SEM) 24 h after the seeding. For SEM, the specimens were fixed with 4% formaldehyde for 24 h and washed three times with PBS to remove unattached cells. Specimens were dehydrated by rinsing with gradually increased ethanol. Afterward, ethanol was exchanged by hexamethyldisilazane (HMDS, Sigma-Aldrich, St. Louis, MO, USA), the specimens were coated with gold and observed under the scanning electron microscope (SEM; JEOL-JSM IT 300, JEOL, Tokyo, Japan) at an accelerating voltage of 15 kV. The SEM images of cross-sectional and surface views were acquired. SEM analysis was performed in triplicates for each type of preparation.

### 2.6. Focal Adhesion Staining

Cell staining using an actin cytoskeleton and focal adhesion staining kit (Catalog Number FAK100, Millipore, Burlington, MA, USA) was performed after 2, 7, and 14 days of culture following the instructions provided by the manufacturer [38]. Microscopy was performed with a fluorescent microscope (Revolve4 RVL-100-G, Echo, San Diego, CA, USA).

### 2.7. Gene Expression Analysis by qPCR

The expression of various genes involved in the focal adhesion was analyzed 2, 7, and 14 days of culture by qPCR similarly to previously described methods. Cell lysis and reverse transcription were done using Cells-to-CT Bulk Lysis Reagents and (Invitrogen, Carlsbad, CA, USA) Cells-to-CT Bulk RT reagent (Ambion/Applied Biosystems, Foster City, CA, USA), respectively [39,40]. qPCR was performed in ABI StepOnePlus device (Applied Biosystems, Foster City, CA, USA) using the Taqman gene expression assays (Applied Biosystems, Foster City, CA, USA) with the following ID numbers: GAPDH, Hs99999905_m1; focal adhesion kinase (FAK), Hs00169444_m1; α-smooth muscle actin (α-SMA), Hs00909449_m1 and the integrin subunits β1 (ITGβ1), Hs01127536_m1; α4 (ITGα4), Hs00168433_m1; β4 (ITG-β4), Hs00173995_m1; α5 (ITGα5), Hs01547673_m1; α6 (ITGα6), Hs01041011_m1. qRT-PCR reactions were performed in 96-well plates using the following thermocycling conditions: 95 °C for 10 min, 50 cycles, each for 15 s at 95 °C and at 60 °C for 60 s. The point at which the PCR product was first detected above a fixed threshold (termed cycle threshold, Ct) was determined for each sample. Changes in the expression of target genes were calculated by a 2^−ΔΔCt^ method using the following formula: ΔΔCt = (Ct^target^ − Ct^GAPDH^) sample − (Ct^target^ − Ct^GAPDH^) control.

### 2.8. Statistical Analysis

Differences between implant surfaces were assessed using the Friedman Test followed by the Kruskal-Wallis test for the pairwise comparisons. *p*-values of less than 0.05 were considered statistically significant. Statistical analysis was performed using SPSS 24.0 software (IBM, Armonk, NY, USA). Data are presented as mean ± SEM of eight different experiments performed with hGFs isolated from eight different donors. Each experiment was performed in technical duplicates.

## 3. Results

### 3.1. Proliferation/Viability of hGFs Grown on ZrM, ZLA, TiM, SLA, and TCP

Proliferation/viability of primary hGFs grown on ZrM, ZLA, TiM, SLA, and TCP group after 2 and 7 days of culture is presented in Figure 1. Proliferation/viability of hGFs grown on all surfaces was gradually increased with prolonged culture time. At 2 days, proliferation/viability of hGFs grown on smooth surfaces was significantly higher than moderately rough surfaces of similar material, i.e., ZrM vs. ZLA and TiM vs. SLA (*p* < 0.05), whereas at 7 days, no significant effect of roughness was observed. At all investigated time points (2 and 7 days), no statistically significant difference in proliferation/viability was observed between hGFs grown on surfaces of different materials and comparable roughness, i.e., ZrM vs. TiM and ZLA vs. SLA.

### 3.2. Scanning Electron Microscopy Analysis (SEM)

The exemplary SEM images of hGFs attached to the different surfaces 24 h after the seeding are shown in Figure 2. Cell shape was drastically different between machined and moderately rough surfaces. On both machined surfaces, hGFs were attached along the grooves remaining after machine treatments. On moderately rough surfaces, hGFs shape was less prolonged, with more attachment points to the different surface structural features.

### 3.3. Fluorescence Microscopy

Figure 3 shows focal adhesion staining of hGFs grown on the different surfaces for 2, 7, and 14 days. Substantial differences in the cell shape between smooth machined and moderately rough surfaces were observed throughout the whole culture period: hGFs grown on smooth surfaces were prolonged and arranged along one axis, whereas those grown on moderately rough surfaces were randomly attached to the surface.

### 3.4. Gene Expression Analysis

Gene expression of FAK and α-SMA in hGFs grown on different implant surfaces is shown in Figure 4. Within surfaces of similar material, significantly higher FAK expression levels were observed on moderately rough SLA than on smooth TiM after 2 and 14 days, and α-SMA gene expression was higher for SLA than TiM surface after 14 days. No significant differences in FAK and α-SMA gene expression were observed between ZrM and ZLA surfaces. Within surfaces of comparable roughness, FAK gene expression was significantly higher for moderately rough titanium SLA surface than for moderately rough zirconium ZLA surface after 2 and 14 days. The gene expression of α-SMA was similar in hGFs grown on ZLA and SLA surfaces. No significant difference in FAK and α-SMA expression was between machined surfaces.

Gene expression of various integrin subunits in hGFs grown on different implant surfaces are shown in Figure 5. Within surfaces of similar material, the gene expression of ITG-β4, ITG-α5, and ITG-α6 was generally higher in hGFs grown on moderately rough surfaces than on smooth surfaces. For ITG-β4, significant differences were observed for ZrM vs. ZLA after 7 days and TiM vs. SLA after 7 and 14 days. For ITG-α5, significant differences were observed for ZrM vs. ZLA after 7 days and for TiM vs. SLA at all time points. For ITG-α6, significant differences were observed for ZiM vs. ZLA for 2 and 7 days and for TiM vs. SLA for the whole observation period. In contrast, the gene expression of ITG-α4 was significantly higher in hGFs grown on smooth ZrM than on moderately rough ZLA after 14 days; otherwise, no effect of roughness was found. The gene expression of ITG-β1 was not affected by surface roughness. Within surfaces of comparable roughness, the gene expression ITG-α5 and ITG-α6 were higher in cells grown on moderately rough Ti/SLA than on moderately rough Zr/ZLA; significant differences were observed at all time points—except at 7 days for ITG-α5. In contrast, the expression of ITG-β1 was significantly higher in cells grown on smooth ZiM than on smooth TiM surface at all time points and in cells grown on moderately rough Zr/ZLA than SLA surface after 7 days. No significant effect of surface material on the gene expression of ITG-β4 and ITG-α4 was observed.

## 4. Discussion

The process of soft tissue formation is largely driven by hGFs, the most abundant gingival progenitor cells, which play an important role in repairing tissue damage and health maintenance [28,41]. In the present study, we investigated the initial attachment, proliferation, and adhesion of primary hGFs to Zr and Ti surfaces of different roughness to have some hints regarding their potential impact on soft tissue.

The proliferation/viability of hGFs after 2 days of culture was significantly higher on smooth machined ZrM and TiM compared to the corresponding moderately rough ZLA and SLA surfaces. A similar tendency, although without statistical significance, was observed after 7 days of culture. At the same time, no influence of surface material on the proliferation/viability of hGFs was observed. Thus, we can conclude that surface roughness is one of the major parameters influencing cell proliferation, and moderately rough surfaces are associated with lower proliferation. This is in agreement with some previous studies on hGFs showing the inhibitory effect of roughness on hGFs proliferation/viability after 48 to 72 h of culture [34,42]. In contrast, another study did not find any differences in the proliferation/viability of hGFs grown on smooth and moderately rough Ti and Zr surfaces [43]. One of the previous studies also reported the increased hGFs proliferation/viability on smooth Zr surface compared to smooth Ti surface [34], which was not observed by us. The reasons for this difference might be different numbers of hGFs’ donors and various experimental protocols.

We have further observed that the morphology of hGFs is strikingly different between machined and moderately rough surfaces. Cells attached to both machined surfaces were elongated and orient along one axis. In contrast, cells attached to moderately rough ZLA and SLA surfaces had typical fibroblast morphology and were randomly oriented. The morphology and orientation of hGFs on machined surfaces were preserved throughout the whole culture period of 14 days. It may be assumed that hGFs orient along the microscopic grooves formed on ZrM and TiM surfaces through the machining process, seen in the SEM pictures of our previous study [35]. This orientation of hGFs along the groove was already reported by previous studies using machined Ti and TiZr surfaces [42,44]. In another study, the paralleled orientation of hGFs was observed on the polished Ti surface, which was not the case for moderately rough surfaces [33]. Thus, cell orientation on the surface might be manipulated by surface treatment.

We have further focused on the impact of surface material and topography on the expression of several proteins involved in focal adhesion. Particularly, we found that the expression of FAK was significantly higher on moderately rough titanium/SLA surface compared to both moderately rough zirconia/ZLA surface and machined smooth titanium/TiM surface. FAK is a non-receptor protein tyrosine kinase, which is involved in the signal transduction from integrin complexes and activation of intracellular signaling pathways as well as other biological functions such as control of cell motility, proliferation, and migration [30]. It should be noted that it is rather difficult to conclude about the contribution of material and roughness into FAK expression. Data on Ti surface suggest that roughness leads to higher FAK expression. However, no effect of roughness was observed within Zr surfaces. Our data on FAK expression agrees with a previous study, in which a higher FAK expression on sand-blasted rough Ti surface compared to polished Ti surface was reported, but no differences between polished and sand-blasted Zr surface was observed [43]. The higher expression levels of FAK for SLA compared to ZLA might also be explained by different roughness. As shown by our former study, the Sa values for SLA are more than two times higher than those for ZLA (1.16 µm vs. 0.45 µm) [35]. Similarly, a previous study showed that the sand-blasted Ti surface exhibits a Ra value of 0.79 µm, which was markedly higher than that of the sand-blasted Zr surface (0.48 µm) [39]. Therefore, it can be assumed that the expression of FAK is promoted by surfaces with roughness over a certain threshold.

The adhesion capacity of fibroblasts in long-term culture correlates with their level of α-SMA expression and the degree of FA supermaturation [45]. The expression of α-SMA in hGFs was only marginally influenced by surface characteristics. Only after 14 days was the expression of α-SMA for the SLA surface higher compared to the TiM surface. The effect of roughness was specific for the Ti surface: no difference in α-SMA expression was found between ZLA and ZiM surfaces. These data are in line with a previous study showing the increase of α-SMA expression on moderately rough Ti surfaces compared to machined and polished smooth Ti surfaces [44]. However, the same study did not observe any difference between TiZr surfaces of different roughness, although a moderately rough TiZr surface exhibited higher roughness than a moderately rough Ti surface. Thus, some specific effects of surface material on the α-SMA expression cannot be excluded. It should also be noted that higher α-SMA expression might also be associated with FAK activation, as reported in a previous study on fibroblasts [32].

Integrins are transmembrane proteins, which interact with the extracellular matrix through their extracellular domains and activate numerous signaling pathways through their intracellular domains [46]. The effect of surface roughness and material seems to be specific for various integrin subunits. For one group of integrin subunits, which includes ITG-β4, ITG-α5, and ITG-α6, a more or less pronounced stimulating effect of surface roughness on the expression levels of these subunits was observed. Moreover, significantly higher expression of ITG-α5 and ITG-α6 was observed on moderately rough Ti compared to moderately rough Zr surface. However, these differences might be related to the different roughness of SLA and ZLA surfaces. Since we did not observe any difference in the gene expression of these integrin subunits between machined ZrM and TiM, we can conclude that their expression is regulated mainly by surface roughness. An increased expression of ITG-α5 is in agreement with its mutual role as the sensor of surface microscale features, as suggested by a previous study on osteoblasts [47].

The gene expression of two other investigated integrin subunits, namely ITG-β1 and ITG-α4, exhibited a strikingly different dependency on surface characteristics. Their expression tended to be higher on Zr compared to Ti surfaces. Moreover, an inhibitory effect of surface roughness on their expression was sometimes observed. The various dependency of integrin subunits expression on different surfaces might be associated with some functional role, which should be further clarified.

The data of the present study suggests an interrelationship between cell morphology and the expression of genes involved in the FA and attachment. Particularly, moderately rough surfaces promote a random attachment of cells to the surface and FAK expression and certain integrin subunits. Although the relationship between the expression of integrin subunits and cellular signaling is not well established, one can assume that moderately rough surfaces might promote particular cellular signaling through specific integrins. It can be further concluded that moderately rough surfaces promote the formation of tissue with homogeneous anisotropic properties. In contrast, a machined surface could be useful if one would be interested in the formation of isotropic tissue with the fibers oriented in one direction. The shape of the attached cells might also determine the tissue formation process, as it was proved to be an essential factor influencing the cell differentiation fate without any other external factor [48].

Our data on the minor effect of the surface material on the hGFs response corresponds to clinical data showing no differences in the soft tissue health seen in peri-implant mucosa adjacent to zirconia and titanium abutment surfaces [49,50,51]. However, Bienz et al. state lower plaque and bleeding scores for zirconia implants under experimental mucositis conditions [50], and another in vivo study suggests blood flow in the tissue surrounding zirconia abutments to be similar to that in the soft tissue around natural teeth [52].

The major limitation of our study is in vitro character. We have used only one cell type, whereas the tissue regeneration process is complex and involves various cell types. Moreover, the potential phenotypical differences between various donors were not considered. Furthermore, our analysis was limited to the gene expression; protein expression and the phosphorylation status of FAK should be further considered. Finally, as mentioned above, the evaluation of the role of roughness and material in hGFs response is aggravated by different roughness of SLA and ZLA surfaces.

## 5. Conclusions

In conclusion, our results suggest that surface topography is an essential determinant for cell response and presumably soft tissue formation compared to the surface material. Some differences between different materials can be explained by the fact that similar treatment of the surfaces with different materials results in different topography due to various intrinsic properties of materials. Our data also imply some roughness threshold, which is essential for promoting soft tissue healing, but the optimal roughness is still to be determined.

## Figures and Tables

**Figure 1 biology-10-00356-f001:**
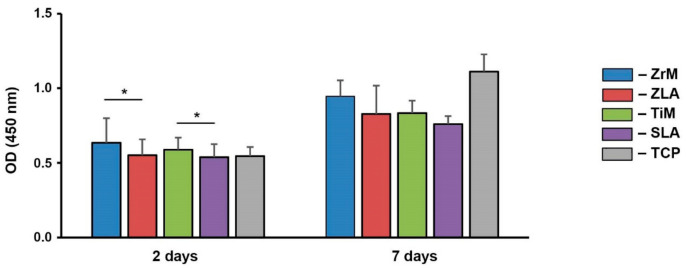
Proliferation/viability of primary human gingival fibroblasts grown on different surfaces. hGFs were cultured on smooth machined zirconia (ZrM), sand-blasted acid-etched moderately rough zirconia (ZLA), smooth machined titanium (TiM), sand-blasted acid-etched moderately rough titanium (SLA) surfaces, and tissue culture plastic (TCP). Cell proliferation/viability was measured by CCK-8 method after 2 and 7 days of culture. *Y*-axis shows the optical density values measured at 450 nm. Data are presented as mean ± SEM of 8 independent experiments with cells isolated from 8 different healthy donors. *—significant difference between the groups (*p* ≤ 0.05).

**Figure 2 biology-10-00356-f002:**
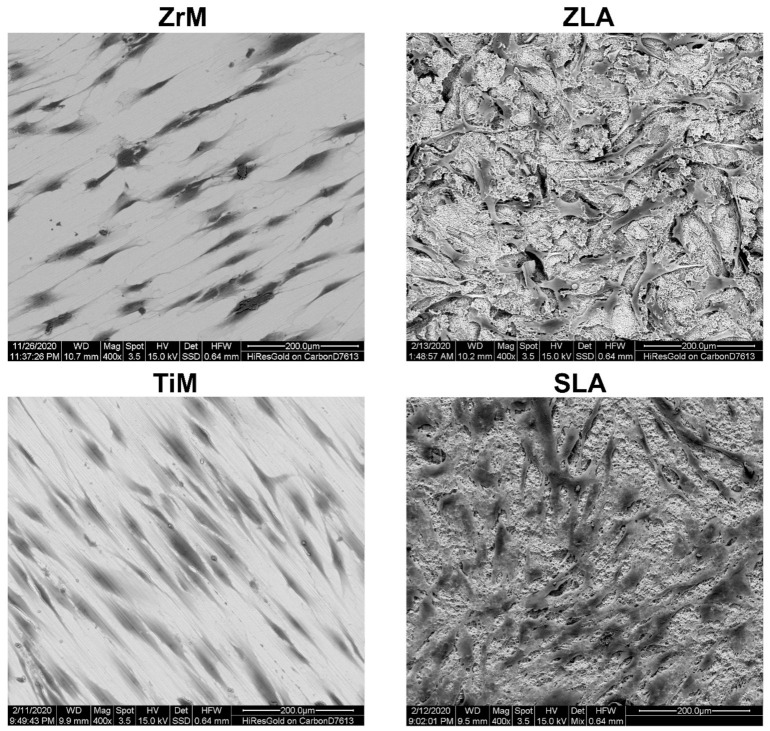
Scanning electron microscopy analysis of hGFs grown on ZrM, ZLA, TiM and SLA. hGFs after 2 days of culture. Scale bar = 200 μm.

**Figure 3 biology-10-00356-f003:**
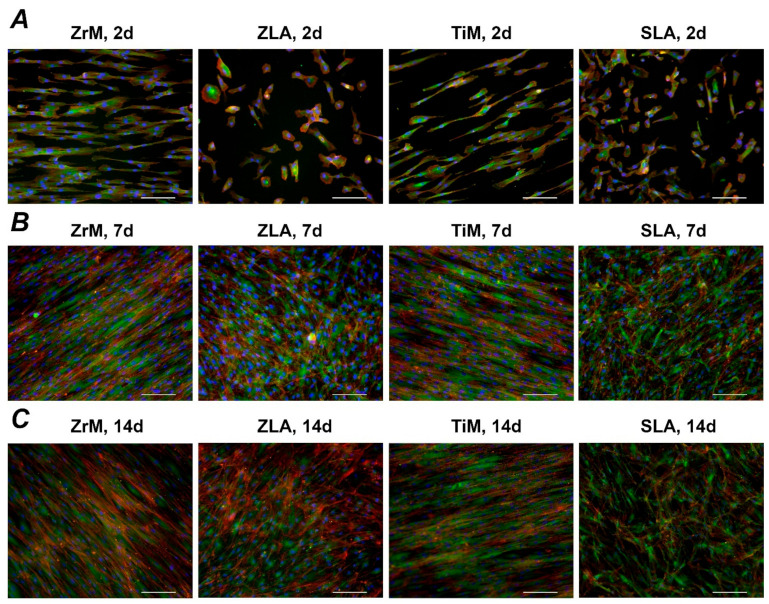
Fluorescence microscopy images of primary hGFs cultured on a smooth machined ZrO_2_ surface (ZrM); a moderately rough ZrO_2_ surface (ZLA); smooth machined Ti surface (TiM); a moderately rough, sand-blasted, large grain acid-etched Ti surface (SLA). Cell culture was performed for 2 (**A**), 7 (**B**), and 14 (**C**) days; F-actin was stained with TRITC-conjugated Phalloidin (red), focal adhesions with anti-Vinculin visualized by fluorescein isothiocyanate (FITC) (green), and the nucleus with 40,6-Diamidin-2-phenylindol (DAPI) (blue). Scale bars correspond to 100 µm.

**Figure 4 biology-10-00356-f004:**
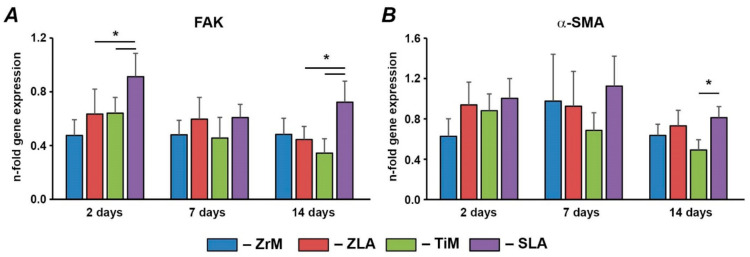
Gene expression of FAK and α-SMA in hGFs grown on different surfaces. hGFs were seeded on one of the following surfaces: smooth zirconia machined surface (ZrM), moderately rough zirconia surface (ZLA), smooth machined titanium surface (TiM), or moderately rough titanium surface (SLA). Expression of (**A**) FAK and (**B**) α-SMA in hGFs grown on ZrM, ZLA, TiM and SLA at 2, 7, and 14 days. *Y*-axis represents the *n*-fold expression levels of the target gene in relation to the cells in the TCP group (control) calculated by the 2^−ΔΔCt^ method. Data are presented as mean ± SEM of 8 independent experiments with hGFs of 8 different donors (*n* = 8), * significant difference between groups as tested by Wilcoxon test (*p* < 0.05).

**Figure 5 biology-10-00356-f005:**
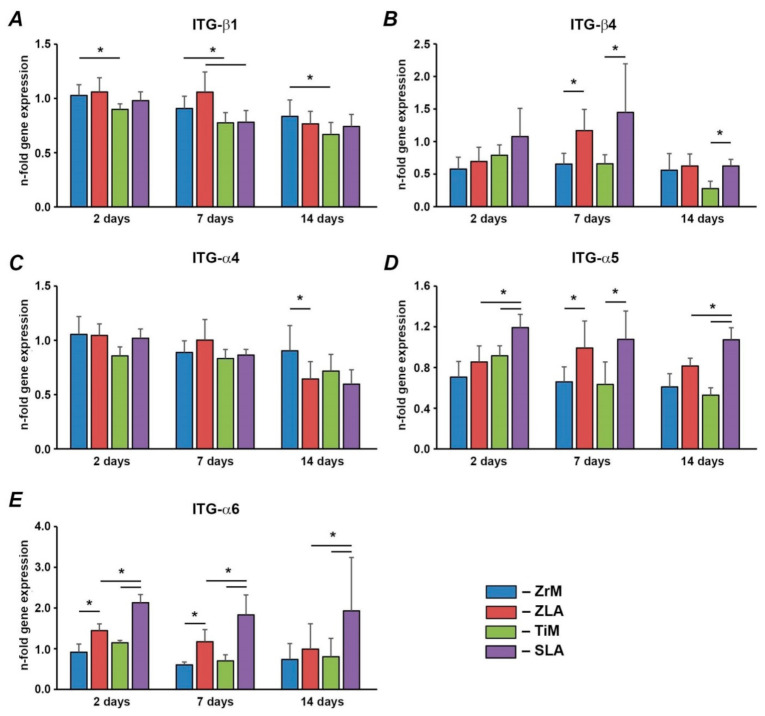
hGFs were seeded on one of the following surfaces: smooth zirconia machined surface (ZrM), moderately rough zirconia surface (ZLA), smooth machined titanium surface (TiM), or moderately rough titanium surface (SLA). Expression of (**A**) ITG-β1, (**B**) ITG-α4, (**C**) ITG-β4, (**D**) ITG-α5, and (**E**) ITG-α6 in hGFs grown on ZrM, ZLA, TiM, and SLA at 2, 7, and 14 days. *Y*-axis represents the *n*-fold expression levels of the target gene in relation to cells in TCP group (control) calculated by the 2^−ΔΔCt^ method. Data are presented as mean ± SEM of 8 independent experiments with hGFs of 8 different donors (*n* = 8). *—significant difference between groups as tested by Wilcoxon test (*p* < 0.05).

## Data Availability

The data presented in this study are available on request from the corresponding author.

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
