# Peer review of "Impact of Implant Surface Material and Microscale Roughness on the Initial Attachment and Proliferation of Primary Human Gingival Fibroblasts"

_biology, 2021, doi:10.3390/biology10050356_

Round 1
Reviewer 1 Report
I have accepted the task to review the quite interesting paper titled “Impact of Implant Surface Material and Microscale Roughness on the Initial Attachment and Proliferation of Primary Human Gingival Fibroblasts” by Rausch et al.. From my personal experience and professional curiosity, I find papers communicating discoveries relating to implants in general very important. The Authors focused on the comparison between Zr and Ti based implants, machined and acid-etched, and their effect on hGFs.
As a Reviewer, I took the privilege of pointing out the following issues:
Lines 21-22: “particularly their topography, on soft tissue regeneration.” - this sentence is incomplete.
Line 46: “. The success of implant therapy is largely osseointegration,” – dependent on ?
Lines 48-50: “Despite high implant success and survival rates [4], a satisfactory aesthetic and functional outcome that is equivalent to natural and healthy teeth is still a challenge in implant dentistry.” This claim is quite odd. Aesthetics has been already achieved, functionality per se as well. The problem results in the implantation.
Lines 74-67: “Additionally, surface hydrophilicity and nanoscale roughness are considered as important parameters influencing osseointegration [24].” It is worth mentioning surface functionalities: peptides, proteins used for implant surface functionalization, e.g. Jurczak et al. doi: 10.1016/j.cis.2019.102083.
Line 140: “a DMEM” – missing e.g. “fresh portion of” ?
Lines 315-324: This part of the Discussion seems redundant, can be easily condensed so the Authors can move on to the main points of this part.
The Discussion part is written without the necessary background, i.e. Authors do not present their result in the context of already published work and/or known hypotheses, from which this publication could only benefit. Some results are mentioned, however for an initial conclusion and a comparison/confirmation, the reader must backtrack the references.
Author Response
GENERAL COMMENT
I have accepted the task to review the quite interesting paper titled “Impact of Implant Surface Material and Microscale Roughness on the Initial Attachment and Proliferation of Primary Human Gingival Fibroblasts” by Rausch et al.. From my personal experience and professional curiosity, I find papers communicating discoveries relating to implants in general very important. The Authors focused on the comparison between Zr and Ti based implants, machined and acid-etched, and their effect on hGFs.
As a Reviewer, I took the privilege of pointing out the following issues
RESPONSE
Thank you for your feedback and the overall positive evaluation.
COMMENT 1
Lines 21-22: “particularly their topography, on soft tissue regeneration.” - this sentence is incomplete.
RESPONSE
Thank you for mentioning this point; the sentence was improved (line 35).
COMMENT 2
Line 46: “. The success of implant therapy is largely osseointegration,” – dependent on ?
RESPONSE
The sentence was improved (line 62).
COMMENT 3
Lines 48-50: “Despite high implant success and survival rates [4], a satisfactory aesthetic and functional outcome that is equivalent to natural and healthy teeth is still a challenge in implant dentistry.” This claim is quite odd. Aesthetics has been already achieved, functionality per se as well. The problem results in the implantation.
RESPONSE
We removed this sentence.
COMMENT 4
Lines 74-67: “Additionally, surface hydrophilicity and nanoscale roughness are considered as important parameters influencing osseointegration [24].” It is worth mentioning surface functionalities: peptides, proteins used for implant surface functionalization, e.g. Jurczak et al. doi: 10.1016/j.cis.2019.102083.
RESPONSE
Thank you for advising this paper; we have referenced it in the revised version (lines 88-90).
COMMENT 5
Line 140: “a DMEM” – missing e.g. “fresh portion of” ?
RESPONSE
The sentence was improved (line 153).
COMMENT 6
Lines 315-324: This part of the Discussion seems redundant, can be easily condensed so the Authors can move on to the main points of this part.
RESPONSE
This paragraph was condensed as suggested by the Reviewer (lines 330-333).
COMMENT 7
The Discussion part is written without the necessary background, i.e. Authors do not present their result in the context of already published work and/or known hypotheses, from which this publication could only benefit. Some results are mentioned, however for an initial conclusion and a comparison/confirmation, the reader must backtrack the references.
RESPONSE
Thank you for your criticism; we have improved the discussion section and discussed our data in the context of previously published studies more deeply. The revised version are highlighted in yellow (lines 339-347; 356-360; 368-380; 386-391).
Reviewer 2 Report
This in vitro study aimed to assess the impact of surface characteristics on human gingival fibroblasts attachment, adhesion, proliferation and gene expression. The samples included titanium and zirconia discs, respectively with a machined and a moderately rough surface, respectively. The study results confirm that surface roughness is the most important surface characteristic affecting cell response. Less impact is attributable to the bulk material.
Zirconia is actually an extraordinary material with many potential applications. Zirconia implants have already become a reality with an elective indication in esthetic areas. However, the number of the research studies and knowledge on Zr implants is still extremely smaller than those on Ti implants. The paper is then of interest to the scientific community and suitable for the publication after minor revision.
Abstract
- Line 22: the term “regeneration” is not appropriate in this context. Substitute with “integration” or “healing”.
Introduction
- Line 46: please check for missing words?
- Line 51: please eliminate the term “regeneration”. Peri-implant soft tissue healing cannot be considered a “regeneration”.
- Line 60-62: the authors should mention the novel techniques used to obtained chemically modified or coated surfaces (PMID: 26800182).
- Line 67: see comments above
Materials and methods
- Line 207: “Data are presented as mean ± s.e.m. of eight different experiments performed with hGFs isolated from eight different donors.” How many discs were tested in each experiment? The number of discs tested in each analysis should be specified. Was each experiment of proliferation/gene expression in duplicate/triplicate?
- Line 190-194: is it necessary to enumerate all the ID numbers?
Discussion
- Among study limitations the authors should mention cell phenotypical differences from eight different donors.
- Another limitation is that SLA and ZLA samples did not have the same roughness, as reported by the authors in a previous study. Thus, it is difficult to state whether observed differences are due to the bulk material or to the different roughness.
Minor corrections
- Line 64-65: please check English
- Line147: substitute “ml” with “mL”
- Figure 1 caption: lines 227 to 230 is a repetition of materials and methods. Consider eliminating this section.
- References should be formatted according to the journal authors guidelines
Author Response
Reviewer 2
We are thankful to this Reviewer for the evaluation of the manuscript and generally positive feedback. Below, we provide point-to-point answer to all raised issues.
COMMENT 1
- Line 22: the term “regeneration” is not appropriate in this context. Substitute with “integration” or “healing”.
RESPONSE
We substituted “regeneration” with “healing” (line 36).
COMMENT 2
- Line 46: please check for missing words?
RESPONSE
Missing word was added. The sentence sound as “The success of implant therapy is largely dependent on osseointegration, the ability of dental implants to fuse with bone, which ensures the load bearing and functionality” (line 62)
COMMENT 3
- Line 51: please eliminate the term “regeneration”. Peri-implant soft tissue healing cannot be considered a “regeneration”.
RESPONSE
The term “regeneration” was substituted with “healing”.
COMMENT 4
- Line 60-62: the authors should mention the novel techniques used to obtained chemically modified or coated surfaces (PMID: 26800182).
- Line 67: see comments above
RESPONSE
Thank you for advising this paper, we have referenced it in the revised version (lines 88-90).
COMMENT 5
- Line 207: “Data are presented as mean ± s.e.m. of eight different experiments performed with hGFs isolated from eight different donors.” How many discs were tested in each experiment? The number of discs tested in each analysis should be specified. Was each experiment of proliferation/gene expression in duplicate/triplicate?
RESPONSE
The experiments were performed in technical duplicates, this information is added to the revised version (lines 221-222).
COMMENT 6
- Line 190-194: is it necessary to enumerate all the ID numbers?
RESPONSE
In the present study, we used commercially available primers (gene expression assays). Due to the privacy policy, we do not have the primers’ sequence. Therefore, we provided the ID numbers to assure the reproducibility of our study by other laboratories.
COMMENT 7
Among study limitations the authors should mention cell phenotypical differences from eight different donors.
Another limitation is that SLA and ZLA samples did not have the same roughness, as reported by the authors in a previous study. Thus, it is difficult to state whether observed differences are due to the bulk material or to the different roughness.
RESPONSE
These issues are mentioned in the revised manuscript as the limitations (lines 433-437).
COMMENT 8
Line 64-65: please check English
RESPONSE
Thank you, the sentence was improved.
COMMENT 9
Line147: substitute “ml” with “mL”
RESPONSE
It is done.
COMMENT 10
Figure 1 caption: lines 227 to 230 is a repetition of materials and methods. Consider eliminating this section.
RESPONSE
We would like do not change this legend, to keep the Figure readable without reference to the main text.
COMMENT 11
References should be formatted according to the journal authors guidelines
RESPONSE
The formatting of the references was improved.
Reviewer 3 Report
the authors made an important job writing and producing this paper. it sounds scientific and it is presented in a well prepared style. the authors provided an appreciable in vitro study on the expression of fibroblasts' genes when in contact with zirconia and titanium implant. the topic is original and in line with the actual lines of research which investigate the best strategies for oral rehabilitation of tooth loss. the materials and method section and the results are clearly explained and the methods are authorized by proper ethical committee acceptance. I only suggest to improve the manuscript by evidence the aspect of new implant technologies such laser prepared implant collars and surfaces which are actually in commerce and described in literature:
Lollobrigida M, Lamazza L, Capuano C, Formisano G, Serra E, Laurito D, Romanelli M, Molinari A, De Biase A. Physical Profile and Impact of a Calcium-Incorporated Implant Surface on Preosteoblastic Cell Morphologic and Differentiation Parameters: A Comparative Analysis. Int J Oral Maxillofac Implants. 2016 Jan-Feb;31(1):223-31. doi: 10.11607/jomi.4247. PMID: 26800182.
Guarnieri R, Di Nardo D, Gaimari G, Miccoli G, Testarelli L. Short vs. Standard Laser-Microgrooved Implants Supporting Single and Splinted Crowns: A Prospective Study with 3 Years Follow-Up. J Prosthodont. 2019 Feb;28(2):e771-e779. doi: 10.1111/jopr.12959. Epub 2018 Aug 31. PMID: 30168651.
few typos like double spaces were found during the revision of the paper.
Author Response
COMMENT 1
the authors made an important job writing and producing this paper. it sounds scientific and it is presented in a well prepared style. the authors provided an appreciable in vitro study on the expression of fibroblasts' genes when in contact with zirconia and titanium implant. the topic is original and in line with the actual lines of research which investigate the best strategies for oral rehabilitation of tooth loss. the materials and method section and the results are clearly explained and the methods are authorized by proper ethical committee acceptance. I only suggest to improve the manuscript by evidence the aspect of new implant technologies such laser prepared implant collars and surfaces which are actually in commerce and described in literature:
Lollobrigida M, Lamazza L, Capuano C, Formisano G, Serra E, Laurito D, Romanelli M, Molinari A, De Biase A. Physical Profile and Impact of a Calcium-Incorporated Implant Surface on Preosteoblastic Cell Morphologic and Differentiation Parameters: A Comparative Analysis. Int J Oral Maxillofac Implants. 2016 Jan-Feb;31(1):223-31. doi: 10.11607/jomi.4247. PMID: 26800182.
Guarnieri R, Di Nardo D, Gaimari G, Miccoli G, Testarelli L. Short vs. Standard Laser-Microgrooved Implants Supporting Single and Splinted Crowns: A Prospective Study with 3 Years Follow-Up. J Prosthodont. 2019 Feb;28(2):e771-e779. doi: 10.1111/jopr.12959. Epub 2018 Aug 31. PMID: 30168651.
RESPONSE
We are thankful for the overall positive evaluation of our manuscript. Both suggested papers are considered and referenced in the revised version (lines 88-90).
COMMENT 2
few typos like double spaces were found during the revision of the paper.
RESPONSE
The text was carefully checked and the typos were eliminated.